# JAC4 Inhibits EGFR-Driven Lung Adenocarcinoma Growth and Metastasis through CTBP1-Mediated JWA/AMPK/NEDD4L/EGFR Axis

**DOI:** 10.3390/ijms24108794

**Published:** 2023-05-15

**Authors:** Kun Ding, Xuqian Jiang, Zhangding Wang, Lu Zou, Jiahua Cui, Xiong Li, Chuanjun Shu, Aiping Li, Jianwei Zhou

**Affiliations:** 1Department of Molecular Cell Biology & Toxicology, Center for Global Health, School of Public Health, Nanjing Medical University, Nanjing 211166, China; 2Key Laboratory of Modern Toxicology of Ministry of Education, School of Public Health, Nanjing Medical University, Nanjing 211166, China; 3Jiangsu Key Lab of Cancer Biomarkers, Prevention and Treatment, Collaborative Innovation Center for Cancer Medicine, Nanjing Medical University, Nanjing 211166, China; 4Department of Bioinformatics, School of Biomedical Engineering and Informatics, Nanjing Medical University, Nanjing 211166, China

**Keywords:** EGFR, JAC4, JWA, NEDD4L, LUAD, ubiquitination

## Abstract

Lung adenocarcinoma (LUAD) is the most common lung cancer, with high mortality. As a tumor-suppressor gene, JWA plays an important role in blocking pan-tumor progression. JAC4, a small molecular-compound agonist, transcriptionally activates JWA expression both in vivo and in vitro. However, the direct target and the anticancer mechanism of JAC4 in LUAD have not been elucidated. Public transcriptome and proteome data sets were used to analyze the relationship between JWA expression and patient survival in LUAD. The anticancer activities of JAC4 were determined through in vitro and in vivo assays. The molecular mechanism of JAC4 was assessed by Western blot, quantitative real-time PCR (qRT-PCR), immunofluorescence (IF), ubiquitination assay, co-immunoprecipitation, and mass spectrometry (MS). Cellular thermal shift and molecule-docking assays were used for confirmation of the interactions between JAC4/CTBP1 and AMPK/NEDD4L. JWA was downregulated in LUAD tissues. Higher expression of JWA was associated with a better prognosis of LUAD. JAC4 inhibited LUAD cell proliferation and migration in both in-vitro and in-vivo models. Mechanistically, JAC4 increased the stability of NEDD4L through AMPK-mediated phosphorylation at Thr367. The WW domain of NEDD4L, an E3 ubiquitin ligase, interacted with EGFR, thus promoting ubiquitination at K716 and the subsequent degradation of EGFR. Importantly, the combination of JAC4 and AZD9191 synergistically inhibited the growth and metastasis of EGFR-mutant lung cancer in both subcutaneous and orthotopic NSCLC xenografts. Furthermore, direct binding of JAC4 to CTBP1 blocked nuclear translocation of CTBP1 and then removed its transcriptional suppression on the JWA gene. The small-molecule JWA agonist JAC4 plays a therapeutic role in EGFR-driven LUAD growth and metastasis through the CTBP1-mediated JWA/AMPK/NEDD4L/EGFR axis.

## 1. Introduction

Non-small-cell lung cancer (NSCLC) is the most common form of lung cancer, accounting for about 85% of lung cancers, and remains the leading cause of cancer death worldwide, especially lung adenocarcinoma (LUAD) [1]. The targeted therapy and immunotherapy using checkpoint inhibitors have shifted the treatment paradigm for NSCLC [2]. There is no evidence that immunotherapy can prolong the overall survival of patients with stage I to III NSCLC who receive radiotherapy or surgery, and adverse events are more likely to occur in those who receive immunotherapy [3]. In addition, compared with chemotherapy alone, the combined therapy (chemotherapy plus immunotherapy) only extends overall survival (OS) in PDL1-positive patients [4]. Therefore, considering the limited population to benefit from immunotherapy and the potential side effects, targeted therapy has become a new option. In addition, genomics and proteomics have developed rapidly over the past two decades, which provide tools for early diagnosis and target screening of NSCLC [5]. Currently, the five-year survival rate of patients with NSCLC is only 15%. Therefore, there is an urgent need to discover new molecular targets and develop effective therapies to improve the prognosis of patients with NSCLC.

Abnormal activation of the epidermal-growth-factor receptor (EGFR) is due to either its overexpression or mutation, which mediates downstream-signaling overactivation and becomes an important driver of the malignant development of human cancers, including lung, head-and-neck, and brain cancers [6,7]. Targeted therapy via EGFR tyrosine-kinase inhibitors (EGFR-TKI), such as gefitinib or erlotinib, has been widely used in the treatment of patients with EGFR-activating mutations and greatly suppresses the prognosis of patients [8]. However, the vast majority of LUAD patients have acquired resistance to EGFR inhibitors, of which EGFR T790M mutation accounts for about 50% of mutation-resistant patients [9]. Osimertinib (AZD9291) has been developed and successfully used in clinics to overcome EGFR T790M-mediated TKI resistance [10]. Unfortunately, new acquired EGFR-resistant mutations (C797S, G724X, L718X, etc.) to osimertinib inevitably emerge [11,12,13], suggesting that targeting EGFR mutations alone is unlikely to cure EGFR-mutant LUAD patients.

In addition to the inevitable acquired resistance, a large number of NSCLC patients harboring wild-type (WT) EGFR do not respond to TKIs despite EGFR overexpression [14]. Moreover, high expression of wild-type EGFR is not only associated with acquired resistance to third-generation EGFR TKIs but also promotes KARS-driven NSCLC tumorigenesis [15,16]. Numerous studies have shown that the stability of the EGFR protein is a key factor in the development of lung cancer. Even in EGFR-mutation-driven lung adenocarcinoma (LUAD), dysregulated EGFR further accelerates the occurrence and progression of lung cancer [17]. Golgi phosphoprotein 3 (GOLPH3) promotes glioma progression by inhibiting Rab 5-mediated endocytosis and degradation of EGFR [18]. CBLC competes with CBL to bind to EGFR, preventing CBL from degrading EGFR and thus promoting the development of LUAD [19]. These findings suggest that promoting EGFR degradation may be an alternative strategy for lung-cancer therapy.

JWA, a known microtubule-binding protein, was originally cloned from retinoic acid-induced human bronchial epithelial (HBE) cells [20]. The JWA gene has also been identified to quickly respond to and be involved in repairing oxidative-stress-induced DNA damage [21,22]. Clinically, JWA expression at both mRNA and protein levels is frequently downregulated in advanced cancers and is associated with worse survival of patients [23,24,25]. Mechanistically, JWA blocks integrin αvβ3-ILK and MMP2 signaling pathways by degrading SP1 via ubiquitination modifications, thus inhibiting the metastasis and angiogenesis of melanoma and gastric cancer, respectively [26,27,28]. JWA promotes TRAIL-mediated apoptosis in cisplatin-resistant gastric-cancer cells by ubiquitinating DR4 via ubiquitin E3 ligase MARCH 8 [29]. Recently, we identified that the JWA-specific agonist JAC1 suppresses breast-cancer growth via degrading HER2 [30]. Therefore, as a key node molecule, JWA can activate the expression of ubiquitinase through related signaling cascades, thus reducing oncoprotein stability and inhibiting tumor progression. However, whether JWA exerts a similar function on the stability of EGFR in LUAD cells is unknown.

In this study, we provide evidence of the first time that JAC4, an agonist of the JWA gene, suppressed EGFR-driven LUAD growth and metastasis in vivo and in vitro. Mechanistically, JAC4 increased the stability of ubiquitin E3 ligase NEDD4L through AMPK-mediated phosphorylation at Thr367 and further enhanced its activity to accelerate the degradation of either overexpressed or mutant oncogenic EGFR in LUAD cells. More importantly, we found that the combination of JAC4 and osimertinib synergistically inhibited the tumor-bearing growth and metastasis of EGFR-mutant LUAD cells in vivo compared with AZD9291 alone. Finally, we determined that JAC4 bound to transcriptional repressor CTBP1 and removed its suppression on JWA transcription, thus rescuing JWA expression in LUAD cells.

## 2. Results

### 2.1. Higher Expression of JWA Is Associated with Better Prognosis of LUAD

To obtain population data on JWA expression in lung cancer, we analyzed transcriptome and proteome data from lung cancer. Analysis using UALCAN (http://ualcan.path.uab.edu/analysis.html, accessed on 22 November 2021) showed that JWA mRNA expression in lung cancer was significantly lower than that in normal lung tissues in TCGA samples (Figure 1A). Furthermore, by analyzing the Gene Expression Omnibus (GEO), we also found that JWA mRNA expression was higher in normal lung tissues than in paired lung-cancer tissues (GSE19804) and non-paired lung-cancer tissues (GSE19188) (Figure 1B,C). In the lung squamous-cell-carcinoma (LUSC) dataset, lower JWA mRNA expression was detected at stages ranging from moderate dysplasia to invasive carcinoma compared to normal tissue samples (Appendix A). Similarly, we used CPTAC proteomic data to analyze the potential difference of JWA protein expression in different stages of lung adenocarcinoma (LUAD). The data showed that JWA protein expression was significantly reduced in LUAD compared to normal lung tissue, and JWA protein levels gradually decreased with increasing tumor stage in LUAD (Figure 1D,E). Moreover, JWA protein expression was lower in LUSC than in the corresponding normal tissues (Appendix A). Meanwhile, Human Protein Atlas data showed that 65% of lung-cancer patients had low or undetectable expression in 20 lung-cancer samples with JWA staining (Figure 1F,G and Appendix A). Taken together, these results suggest that JWA is downregulated in lung-cancer tissues both in mRNA and in protein levels. In addition, Western blotting showed that JWA protein expression was significantly higher in human bronchial epithelial (HBE) cells than in the non-small lung-cancer cell lines A549, SPCA1, PC9, and H1299 (Figure 1H and Appendix A), which is consistent with JWA mRNA expression in cell lines by qRT-PCR (Figure 1I). Importantly, receiver-operating-characteristic (ROC) analysis indicated that JWA could be used as an independent prognostic biomarker in lung cancer (Figure 1J). Moreover, patients with higher JWA expression had better prognostic outcomes (Figure 1K). The Kaplan–Meier Plotter analysis (http://kmplot.com/analysis/, accessed on 6 January 2022) further showed that patients with high JWA expression had longer times in overall survival (OS), suggesting that high JWA levels are associated with better survival in lung cancer (Appendix A). To investigate the localization of JWA in organs, we assessed the protein expression of JWA across normal human tissues by mass spectrometry from the Human Protein Map (www.humanproteomemap.org, accessed on 20 January 2022), and the data showed that JWA was mainly enriched in lung tissues (Appendix A). Collectively, these results suggest that higher JWA expression is associated with a better prognosis in individuals with lung cancer and that JWA may be a promising potential biomarker of LUAD.

### 2.2. Screening of JWA Small-Molecule Agonist JAC4 for Suppression of LUAD Proliferation and Metastasis

To determine whether the agonists of the JWA gene can selectively activate the expression of JWA in lung-cancer cells, two small-molecular compound agonists, JAC1 [30] and JAC4 [31] (Appendix A), were used in experimental models. Considering the tumor heterogeneity and drug sensitivity, the subcutaneous tumor-bearing model was first constructed to screen for a better tumor-inhibition effect in lung cancer. Based on the tumor-volume and tumor-weight curves, JAC4 showed better anti-proliferation activity than JAC1 (Figure 2A–D). Moreover, Western blotting showed that tumors in the JAC4 treatment groups exhibited higher JWA and BAX protein expression and lower PCNA and BCL2 protein expression (Figure 2E). In addition, immunohistochemistry (IHC) staining showed that decreased Ki67 expression, increased JWA, and cleaved caspase 3 expressions were detected in the JAC4 treatment groups compared to those in control groups (Figure 2F,G). Importantly, JAC4 treatment did not change the body weight of mice (Appendix A), and histological evaluation of the heart, liver, spleen, lung, and kidneys showed no signs of toxicity in the JAC4-treated mice (Appendix A). Moreover, JAC4 treatment did not cause an elevation of serum levels of ALT, AST, BUN, CK, CK-MB, or LDH (Appendix A). Collectively, these findings suggest that JAC4 inhibited lung-cancer progression in vivo, with no obvious toxicity. To investigate whether JAC4 suppresses lung-cancer progression in cells, we treated A549 cells with different concentrations of JAC4. Results show that 10 μM JAC4 treatment for 24 h had the best activation effect on JWA expression (Appendix A). Colony formation and 5-ethynl-2′-deoxyuridine (EDU) staining assays demonstrated that the cells with JAC4 treatment reduced proliferation and growth potential compared to DMSO-treated ones (Figure 2H–K). In contrast, JAC4 treatment had little effect on the growth of normal cells (HBE and BEAS-2B) (Appendix A). Moreover, JAC4 was found to significantly downregulate PCNA protein expression in a dose-dependent manner (Appendix A). In addition, similar to the previous observation that JWA inhibits lung-cancer metastasis [32], transwell cell migration and invasion assays showed that JAC4 treatment greatly inhibited lung-cancer cell migration and invasion compared to DMSO-treated cells (Figure 2L–N and Appendix A). Taken together, these results indicate that JAC4 suppresses lung-cancer cell proliferation and metastasis in vitro. Considering that different chiral molecules may have different effects, we further used a subcutaneous-tumor-bearing model to identify which JAC4 chiral molecule has better antitumor effects (Appendix A). It was found that both the JAC4-R and the JAC4 prototype could significantly inhibit the growth of lung cancer, and JAC4-S had a weak tumor-inhibitory effect; furthermore, JAC4-R did not show a better tumor-suppression effect than the prototype (Appendix A). Therefore, the JAC4 prototype was used in all later models.

### 2.3. JAC4 Promotes EGFR Degradation through the Ubiquitination Proteasome Pathway

We previously reported that JWA suppresses cell migration by negatively regulating HER2 expression in HER2-positive gastric-cancer cells [33]; moreover, JWA agonist JAC1 enhances the ubiquitination of HER2 [30]. HER2 is known as a member of the ERBB family, and EGFR is a vital target in lung cancer; however, the role of JWA and related small-molecule activator JAC4 on EGFR remains unknown. To explore the underlying mechanisms of how JWA affects lung-cancer progression, Western blotting was performed, which showed that the molecular markers of EGFR, P-EGFR, P-AKT, and P-STAT3 (the downstream pathways of EGFR) were downregulated when the cells increased JWA expression; however, these markers were upregulated when JWA was knocked down (Figure 3A). These results indicate that JWA may be a negative regulator of the EGFR pathway in lung cancer. To test whether JAC4 could downregulate EGFR in lung-cancer cells, upon JAC4 treatment, immunoblotting analysis was carried out and showed that the protein expression of JWA was increased, whereas the expression of EGFR was decreased in dose- and time-dependent manners; moreover, the downstream of EGFR-related markers (P-AKT and P-STAT3) were synchronously suppressed (Figure 3B,C and Appendix A). Importantly, the rescue experiments showed that decreased EGFR expression caused by JAC4 treatment was blocked by JWA silence (Figure 3D). These results suggest that the inhibitory effect of JAC4 on EGFR expression is achieved partly through upregulating JWA expression. Meanwhile, lower EGFR protein expression was detected in tumor tissues from the mice receiving JAC4 treatment than the vehicle-control mice (Figure 3E). Moreover, immunofluorescence staining showed that the levels of EGFR were decreased in A549 and SPCA1 cells after JAC4 treatment (Figure 3F and Appendix A). We next examined whether the reduction of EGFR was regulated at either the transcriptional or post-translational level. qRT-PCR analysis showed that the mRNA expression of EGFR was not reduced in JAC4-treated A549 cells (Appendix A). Clinical-patient data from the GEO databases revealed that JWA mRNA expression had no obvious correlation with EGFR expression in cancer (Figure 3G,H). Therefore, JAC4 may regulate the expression of EGFR post-transcriptionally.

To determine whether JAC4 affects the protein stability of EGFR in cells, pulse-chase analysis using cycloheximide (CHX) was carried out and showed that compared with control cells, JAC4 reduced the half-life time of the EGFR protein in both A549 and SPCA1 cells (Figure 3I). Moreover, the JAC4-mediated degradation of EGFR could be efficiently blocked by the proteasome inhibitor MG132 (Appendix A). To further determine whether JAC4 triggered EGFR ubiquitination modification, in-vitro ubiquitination assays were performed in A549 cells with ectopic expression of his-ubiquitin. Results show that ubiquitinated EGFR was increased in cells after JAC4 treatment (Figure 3J). Furthermore, enhanced EGFR ubiquitination and reduced EGFR protein levels were observed in JWA-overexpressed A549 cells; conversely, knockdown of JWA reduced EGFR ubiquitination and increased EGFR expression (Appendix A). Collectively, these findings suggest that JAC4 may trigger ubiquitin–proteasomal degradation of EGFR in lung-cancer cells via upregulating the expression of JWA.

### 2.4. JAC4 Ubiquitinates EGFR by E3 Ubiquitin Ligase NEDD4L

To identify which E3 ubiquitin ligases were involved in process of EGFR ubiquitination, we first used the UbiBrowser database to predict the candidates from the human-ubiquitin-ligase (E3) substrate interactions [34]. The top 20 E3 ubiquitin ligases were found to be potentially involved in EGFR ubiquitination (Figure 4A and Appendix A). Next, we further investigated the expression of E3 ligases by mining 60 pairs of matched lung-cancer microarray data (Figure 4B). The mRNA level of NEDD4L but not MIB1, WWP1, or other ligases was downregulated in patients with lung cancer. These results helped us further narrow down the candidate cope of E3 ligases. Moreover, immunoblot data showed that among the top five E3 ubiquitin ligases (based on confidence scores in the UbiBrowser database), only NEDD4L could be upregulated by JAC4 treatment in human-lung-cancer cells (Figure 4C and Appendix A), which was further confirmed in the JAC4-treated lung-cancer tumor-tissue samples (Figure 4D). We then analyzed the relationship between the NEDD4L levels and the prognosis of lung-cancer patients. Through our analysis of public data regarding the GEO, TCGA, and CPTAC (Clinical Proteomic Tumor Analysis Consortium) databases, the results show that both mRNA and protein levels of NEDD4L were decreased in patients with lung cancer (Appendix A). This result is also consistent with NEDD4L staining by IHC from The Human Protein Atlas (Appendix A). Furthermore, by analyzing the GEO data, we found that patients with high NEDD4L expression had better overall survival, which is consistent with the online Kaplan–Meier Plotter tool (Appendix A). These results suggest that JAC4 can upregulate NEDD4L expression and that NEDD4L is positively associated with a better prognosis for lung-cancer patients.

Substrate binding to ubiquitin ligases is a key event in protein degradation and signal transduction. We analyzed protein interactions between NEDD4L and EGFR. The molecular-docking assay showed that NEDD4L could interact with EGFR physically (Figure 4E). Moreover, we found that ectopically expressed NEDD4L and EGFR proteins interacted with each other in HEK293T cells (Figure 4G,H), and their interaction also occurred at the endogenous level in A549 cells (Figure 4F). To identify which regions in NEDD4L may be bound to EGFR, we constructed a series of truncated plasmids of NEDD4L and transferred them into HEK293T cells (Figure 4I). Co-IP assays revealed that the WW domain was crucial for NEDD4L binding to EGFR (Figure 4J). These data suggest that NEDD4L binds to EGFR mainly through its WW domain.

To further investigate whether NEDD4L reduced EGFR stability, we overexpressed or knocked down NEDD4L in A549 cells. Cycloheximide pulse-chase experiments (CHX-chase) showed that NEDD4L overexpression shortened but NEDD4L knockdown prolonged EGFR half-life time in A549 cells (Figure 4K,L). Next, we found that NEDD4L overexpression significantly increased the ubiquitination of EGFR (Figure 4M). In contrast, knockdown of NEDD4L reduced the ubiquitination levels of EGFR (Figure 4N). Functionally, both colony formation and transwell assays supported that overexpression of NEDD4L inhibited the proliferation and metastasis of lung-cancer cells, which could be blocked by overexpression of EGFR (Figure 4O,P and Appendix A). In addition, the protein-expression levels of NEDD4L and EGFR were negatively correlated in lung-cancer cell lines (Appendix A). Furthermore, to prove that JAC4 ubiquitinates EGFR by regulating NEDD4L, we performed a CHX-chase assay in siNEDD4L-transfected cells with or without JAC4 treatment. Results show that the enhancement of EGFR protein stability caused by knockdown of NEDD4L was reversed by JAC4 (Appendix A). Our results suggest that JAC4 ubiquitinates EGFR by upregulating E3 ubiquitin ligase NEDD4L expression.

### 2.5. K716 Is Critical for NEDD4L-Mediated Degradation of EGFR by JAC4

To further identify potential lysine residues in EGFR that were ubiquitinated by NEDD4L, we used the phosphosite web tool (http://www.phosphosite.org/, accessed on 17 January 2022) to conduct a prediction assay. Results showed that there were seven ubiquitination sites (K716, K737, K754, K860, K867, K929, and K970) in the intracellular domain of EGFR (Figure 5A). We next generated a panel of EGFR mutants by replacing individual lysine (K) residues with arginine (R) and tested their responses to CHX treatment. We found that both K716R and all mutants were resistant to CHX treatment (Figure 5B). Compared with WT EGFR, the K716R mutation clearly increased the protein stability of EGFR in A549 and SPCA1 cells (Figure 5C–E). We further performed in-vitro ubiquitination assays to determine whether the mutations affect EGFR ubiquitination levels. Compared with WT EGFR, the K716R mutation significantly reduced the ubiquitination ability of EGFR (Figure 5F,G). We found that JAC4 failed to downregulate EGFR protein levels in cells that were transfected with Flag-EGFR K716R, indicating that K716 was the actual ubiquitination site targeted by JAC4 (Figure 5H). Moreover, both colony formation, EDU incorporation, and transwell assays supported the finding that the K716R mutation had higher proliferative and metastatic potential than the EGFR WT (Figure 5I–M and Appendix A). These results suggest that K716 of EGFR is critical for NEDD4L-mediated ubiquitination and degradation of EGFR by JAC4.

### 2.6. JAC4 Suppresses EGFR T790M-Driven LUAD Growth and Metastasis

NCI-H1975 (EGFR L858R/T790M) is a LUAD cell line that is known to be resistant to first- and second-generation EGFR inhibitors. Since the mutations of EGFR in NCI-H1975 did not overlap with ubiquitination site K716, we investigated whether JAC4 also exerted a similar effect on it. Through a colony-formation assay, we found that JAC4 significantly inhibited the growth of NCI-H1975 cells (Figure 6A,B). To further confirm whether JAC4 reduced EGFR protein stability, we treated the NCI-H1975 cells with CHX. Notably, JAC4 treatment reduced the half-life time of the endogenous EGFR protein (Figure 6C,D). In addition, JAC4 enhanced ubiquitination to degrade EGFR in in-vitro ubiquitination experiments (Figure 6E). Consistent with previous findings, the half-life of the transfected EGFR-K716 mutant was longer than that of the EGFR WT in NCI-H1975 cells (Figure 6F,G). The above results indicate that JAC4 inhibited EGFR-T790M cell growth in vitro by degrading EGFR.

To determine the effect of JAC4 in EGFR-mutant LUAD progression in vivo, we created a subcutaneous tumor-bearing mouse model. Once the average volume of the tumors reached 100 mm^3^, the animals were randomly assigned to receive vehicle, AZD9291, and JAC4 in combination with AZD9291 and JAC4 (Figure 6H). Data show that JAC4 treatment inhibited EGFR-T790M-driven LUAD growth and that the combined treatment of AZD9291 and JAC4 synergistically suppressed LUAD growth compared with that in mice treated with AZD9291 (Figure 6I–K). Moreover, the expression of both JWA and NEDD4L was increased; however, EGFR, p-EGFR, p-AKT, and p-STAT3 expression levels were reduced in JAC4-treated tumor tissues compared to those in the control group (Figure 6L and Appendix A). Likewise, we also observed higher NEDD4L expression and lower tumor-cell proliferation (Ki67 staining) using immunohistochemistry (IHC) (Figure 6M and Appendix A). Importantly, we did not observe obvious weight loss in mice with either JAC4 alone or the combined treatment (Appendix A). Besides, the representative serum biomarkers, including ALT, AST, BUN, CK-MB, and CK, and the histological morphology of the heart, liver, spleen, lung, and kidneys, revealed no apparent changes compared with those in the control group (Appendix A).

To evaluate the effect of JAC4 on the proliferation and metastasis in EGFR-mutant cells, we constructed an in-vivo lung-metastasis model by tail-vein injection of NCI-H1975 cells (Figure 6N). The results show that JAC4 significantly inhibited lung metastasis, and the combination of JAC4 with AZD9291 synergistically suppressed lung-cancer metastasis (Figure 6O and Appendix A). The H&E-staining data indicate that the relative metastasis burden in the JAC4-treated group was significantly reduced compared to that of the control group (Figure 6P,Q). Concurrent with these findings, analysis of public clinical data recorded in the TCGA and GEO databases demonstrated that EGFR-mutant lung-cancer patients whose tumors expressed higher JWA expression were associated with better overall survival (Figure 6R).

### 2.7. JWA/AMPK Axis Stabilizes NEDD4L Expression by Phosphorylating NEDD4L at Thr367

How does JAC4 activate NEDD4L in NSCLC cells? Previous studies have shown that JWA suppresses pancreatic-cancer progression via the AMPK-FOXO3a axis [35]. Moreover, deficiency in GTRAP3-18 (the homologous protein of JWA) in mice results in AMPK inhibition [36]. In addition, NEDD4L protein stability is known to be dependent upon its phosphorylation modification [37]. Thus, we speculated that JWA may stabilize E3 ubiquitin ligase NEDD4L through an AMPK-mediated phosphorylation-signaling cascade. To confirm this hypothesis, we first determined the potential positive correlation between JWA expression and P-AMPK level by analyzing the TCGA-RPPA (reverse-phase protein array) and CPTAC-phosphoproteome in lung-cancer patients (Appendix A). Upon further analysis of the CCLE (Cancer Cell Line Encyclopedia)-RPPA, a negative correlation between P-AMPK and EGFR protein expression was found in lung-cancer cell lines (Appendix A). In addition, RNA-seq analyses of AMPK-KO MEF cells revealed higher EGFR expression compared to that in AMPK-WT cells in the public database (Appendix A), which supported a negative correlation between AMPK and EGFR expression. Furthermore, NEDD4L expression was insensitive to JAC4 treatment when knocking down AMPK or using an AMPK inhibitor (Figure 7A and Appendix A). Subsequently, transwell and EDU assays also confirmed that the effect of JAC4 on lung-cancer proliferation and migration could be inhibited when knocking down AMPK or using an AMPK inhibitor (Figure 7B–E). Consistently, the protein expression of P-AMPK and NEDD4L were elevated in NCI-H1975 and SPCA1 tumor samples upon JAC4 treatment (Figure 7F–I). To demonstrate whether Foxo3 expression is involved in the regulation of NEDD4L expression, we constructed the si-Foxo3 A549 cell line, and the data showed that the expression of NEDD4L did not change in si-Foxo3 A549 cells (Appendix A). Therefore, the JWA/AMPK axis enhanced NEDD4L stability in a Foxo3-independent manner. Our results were also supported by another study that showed that AMPK phosphorylates NEDD4L in Xenopus and is critical for NEDD4L stability [38]. To further investigate the interaction between AMPK and NEDD4L, we completed an AMPK-NEDD4L docking assay, and the results indicate that AMPK interacted with NEDD4L physically (Figure 7J). In addition, the activated form of AMPK (P-AMPK T172) was co-immunoprecipitated with NEDD4L (Figure 7K). To determine AMPK phosphorylation sites on NEDD4L, the Group-based Prediction System 3.0 (GPS 3.0) software and phosphoNET kinase predictor were used (Figure 7L), and the data show that T367 in NEDD4L was a potential phosphorylation site of AMPK. Human NEDD4L T367 was a highly conserved locus across different species (Figure 7M). To investigate whether NEDD4L T367 phosphorylation is modified by AMPK, a plasmid encoding NEDD4L WT or a phosphorylation-defective mutant (T367A) was transfected into A549 cells, followed by JAC4 treatment for 24 h and subsequent CHX treatment. The half-life of NEDD4L with the T367A mutation was shortened compared with that of NEDD4L WT (Figure 7N). These findings suggest that JAC4 enhanced the stability of NEDD4L via AMPK-triggered phosphorylation at Thr367.

### 2.8. JAC4 Upregulates JWA Expression by Binding to Its Transcriptional Suppressor CTBP1 in NSCLC Cells

Since it was determined that JAC1 promotes JWA transcription by binding toYY1 [39], we speculate that JAC4 increased JWA transcriptional activity by affecting JWA–cofactor interaction. To confirm this hypothesis, both A549 and HBE cells were treated with biotin-JAC4 for 24 h; compared with proteins interacting with biotin alone, 44 potential JAC4-interacting proteins were identified by streptavidin-immunoprecipitation assay and mass-spectrometry analysis (MS) (Figure 8A). We further compared these potential proteins with the predicted JWA-promoter-specific transcription factors in public databases (http://jaspar.genereg.net/, accessed on 19 August 2022). As a result, five transcriptional suppressors, including BCLAF1, CTBP1, ZMYM3, SMC3, and HDAC2, were obtained (Figure 8B). To further confirm which one mediated the role in JAC4 upregulating JWA expression, we completed siRNA-transfection assays to reduce the relevant transcriptional factors and measure JWA expression. The data show that only inhibition of CTBP1 could upregulate JWA expression (Appendix A), which is consistent with the verified data indicating that overexpression of CTBP1 resulted in a reduction of JWA in HBE cells (Figure 8C). Moreover, we generated CTBP1-knockout A549 cells through the Cas9 gene-editing technique (Figure 8D and Appendix A). As expected, JWA expression was obviously elevated in CTBP1-knockout A549 cells (Figure 8E). Subsequently, we investigated whether JAC4 activating JWA expression was due to inhibition of CTBP1. The data show that CTBP1 protein levels were unaffected by JAC4 treatment (Appendix A). Therefore, JAC4 may interact with CTBP1 and block its transcriptional-repressor function on the JWA gene in both HBE and lung-cancer cells. Molecular-docking and Co-IP assays also verified that JAC4 could interact with CTBP1 (Figure 8F,G).

To further confirm the interaction between JAC4 and CTBP1 in cells, we conducted thermal-shift assays (CETSA) and found that JAC4 treatment led to significant thermal stabilization of CTBP1 (Figure 8H). These results indicate that JAC4 directly interacts with CTBP1. As a transcriptional suppressor, nuclear translocation is necessary for CTBP1 post-activation. Our data show that JAC4 treatment reduced its nuclear levels and increased cytoplasm levels of CTBP1 in A549 cells in a dose-dependent manner, suggesting that JAC4 also reduced CTBP1 translocation (Appendix A). Moreover, when CTBP1 was knocked out, its transcriptional suppression on JWA expression was completely removed; therefore, JAC4 was unable to exert subsequent inhibition of lung-cancer growth and metastasis via the CTBP1-JWA pathway in these CTBP1-deficient cells (Figure 8I–L and Appendix A). Contrary to lower JWA expression in lung cancer, CTBP1 was highly expressed in A549, NCI-H1975, and HCC827 lung-cancer cell lines (Figure 8M) and lung-cancer tissues compared to corresponding controls (Figure 8N). Therefore, a negative correlation between CTBP1 and JWA expression was identified in human-lung-cancer tissues (Figure 8O). These results suggest that CTBP1 might be a valuable target involved in the progression of NSCLC due to its role as a transcriptional suppressor, and that JAC4 targeted CTBP1 and therefore rescued the expression of tumor suppressors like JWA.

## 3. Discussion

Gene amplification and mutations of EGFR have been implicated in the pathogenesis and progression of many malignancies, including lung cancer [40]. EGFR tyrosine-kinase inhibitors (TKIs) have been widely used in clinical practice, significantly prolonging the survival of patients [41]. However, up to 60% of patients eventually develop resistance within 10–14 months after first-generation TKIs [42]. Furthermore, many EGFR-mutant patients are insensitive to EGFR TKIs, and patients with EGFR WT are insensitive to TKIs despite overexpression [14]. Recent studies have shown that EGFR stability is a fundamental mechanism for maintaining the homeostasis of EGFR signaling. Therefore, targeting and degrading EGFR is a more effective and complete strategy for lung-cancer treatment. Here, based on public lung-cancer transcriptome and proteome data, we found that JWA was lowly expressed in NSCLC and that patients with high JWA expression had a better prognosis. JAC4 significantly activated the expression of JWA in lung-cancer cell lines and tumor tissues, thereby inhibiting the growth and metastasis of EGFR-driven lung cancer in vivo and in vitro. Mechanistically, JAC4 accelerated the degradation of EGFR by activating the phosphorylation of AMPK signaling and stabilizing the expression of NEDD4L at Thr367. The K716 site of EGFR is required for NEDD4L-mediated ubiquitination to degrade EGFR. In addition, JAC4 removed the transcriptional regression of JWA by binding to CTBP1 directly, and therefore rescued normal JWA transcription. Collectively, the small-molecular agonist JAC4 of the JWA gene inhibited EGFR-driven lung-cancer growth and metastasis via the AMPK-NEDD4L-EGFR axis (Figure 9).

Several EGFR-degradation strategies have been reported. One approach is to deliver specific EGFR small-interfering RNAs in vivo; however, the efficacy is limited by rapid degradation and a short half-life in the body [43]. Additionally, protein-hydrolysis-targeted chimeric (PROTAC) technology has been used for EGFR degradation, membrane permeability, solubility, and metabolic stability, which has added to the challenges of synthetic drugs [44]. In this study, we determined that JAC4, an agonist of JWA, can degrade EGFR by activating NEDD4L. Furthermore, since JWA is predominantly localized in the lung and lowly expressed in tumor tissues, this may greatly reduce the toxicity to the normal organism. Although we performed a series of in-vivo and in-vitro functional assays and reversion experiments to confirm that the expression levels of JWA and CTBP1 were necessary for JAC4 to exert its anti-cancer function, we still cannot exclude other downstream molecules that may be involved in the onset of the anti-tumor mechanism of JAC4, which requires further investigation. Interestingly, the previously screened JWA agonist JAC1 could inhibit breast-cancer proliferation by ubiquitinating the expression of HER2 [30], whereas our current study found that JAC4 was superior to JAC1 in terms of tumor suppression in LUAD, which is possibly attributable to the difference in transcription factors necessary for the activation of JWA by the two different compounds: JAC1 activates JWA expression via the transcription factor Yin Yang 1 (YY1) [39], whereas JAC4 promotes JWA expression mainly by reducing the nuclear translocation of CTBP1. Considering the differences, further therapeutic strategies can be selected based on the background expression levels of CTBP1 and YY1 in different cancer types.

NEDD4L belongs to the NEDD4 family of HECT E3 ubiquitin ligases, which includes four WW domains that can specifically recognize targeted substrates containing PPXY, LPXY, or PPR sequences [45]. NEDD4L functions as a tumor-suppressor gene in some cancers, such as breast, pancreatic, and lung cancer [46,47,48]. Our study also found that NEDD4L was negatively associated with proliferation and migration in NSCLC. NEDD4L can also inhibit the expression of various tumor-associated membrane proteins, including LGR5, beta-catenin, and transforming growth-factor beta (TGFβ) receptor [49,50,51]. Although a negative regulatory relationship between NEDD4L and EGFR protein expression has been reported, the specific mechanism has not been elucidated [52]. In addition, inhibition of EGFR-signaling pathway promotes NEDD4L protein expression, which may be caused by its downstream-signaling pathways [52]. Therefore, the deep regulatory relationship between NEDD4L and EGFR, especially on mutant EGFR degradation, has not been elucidated. Our results suggest that the binding of the WW region of NEDD4L to EGFR plays a role in the ubiquitination and degradation of EGFR, thus inhibiting the activation of downstream PI3K/AKT signaling. In addition, mutations or single-nucleotide polymorphisms in the human NEDD4L gene are associated with a variety of diseases [53,54,55]. By analyzing the mutation databases in TCGA and COSMIC, we found that the main mutations of tumor-related NEDD4L are G39V, P197S, S230F, and R253W (Appendix A). These mutations may affect its binding to EGFR and thus fail to degrade EGFR; further studies are required to experimentally validate this hypothesis.

AMPK, a major energy sensor and regulator, regulates cell metabolism and energy homeostasis [56]. AMPK inhibits lung-cancer growth primarily by inhibiting mTORC1 oncogenic signaling [57]. Meanwhile, AMPKα deletion has been shown to promote KRAS-mediated lung-cancer growth and metastasis [58]. Therefore, AMPK has emerged as a potential target for cancer therapy. GTRAP3-18, a homologous protein of JWA, and mice’s loss of GTRAP3-18 resulted in AMPK inhibition [36]. Our study confirmed that JAC4 significantly increased the expression of P-AMPK, and the effect of JAC4 on the growth and metastasis of NSCLC was obviously inhibited after knockdown of AMPK or use of AMPK inhibitor, indicating that JAC4 also acts as an agonist of AMPK to suppress lung-cancer proliferation. NEDD4L expression can be stabilized through ERK and SGK1 phosphorylation cascades [51,59]. We found that AMPK plays a role in phosphorylating and stabilizing NEDD4L in NSCLC, which may be due to the heterogeneity of different cancer types. Unlike JAC4, previous studies have reported that JWA-mimicking peptide JP1 phosphorylates NEDD4L expression via MEK1/2 signaling [60]. Thus, the regulation of NEDD4L by JWA may involve in different mechanisms. Multiple clinical trials have reported that AMPK agonist metformin combined with EGFR-TKIs can prolong the progression-free survival (RFS) in EGFR-mutated NSCLC compared to EGFR-TKIs alone [61,62]. In this study, the combination of JAC4 and osimertinib synergistically inhibited the tumor-bearing growth and metastasis of EGFR-mutant lung-cancer cells in model mice. Therefore, JAC4 has also proven to be potentially valuable in combination with EGFR inhibitors in cancer therapy.

CTBP1 is a critical transcriptional repressor, and deficiency or overexpression of CTBP1 leads to transcriptional imbalances [63]. CTBP1 induces a wide range of tumorigenic- and cancer-stem-cell-relative functions through transcriptional regulation of gene networks [64]. CTBP1 inhibits the adhesion of molecules such as E-cadherin and is associated with the promotion of epithelial–mesenchymal transition (EMT), a step that contributes to the malignant properties of tumors [65]. Interestingly, JAC4 was also found to promote E-cadherin expression, indicating that JAC4 may be involved in the suppression of the EMT process in lung cancer (Appendix A). In addition, several studies have found that CTBP1 forms complexes with multiple epigenetic regulators or transcriptional regressors, and these complexes further recruit epigenetic regulators to control gene expression. CTBP1 also undergoes dynamic post-translational modifications that affect its own stability and subcellular localization [66,67]. Our results show that JAC4 directly bound to CTBP1 and significantly reduced its nuclear overload, thereby restoring JWA transcription levels to their normal state. The thermal stability of CTBP1 was obviously increased after JAC4 treatment, which further confirms that the combination of both may result in changing the properties of CTBP1. The background events and mechanisms, such as the fine mapping of specific amino-acid residues of CTBP1 binding to JAC4, need to be further elucidated.

## 4. Materials and Methods

### 4.1. Cell Culture, Transfection, and Treatment

The HBE and HEK293T cell lines were purchased from the American Type Culture Collection (ATCC, Manassas, VA, USA) and the A549, NCI-H1975, HCC827, SPCA1, PC9, NCI-H1299, H226, H460, and LLC cell lines were purchased from the National Collection of Authenticated Cell Cultures (Shanghai, China). The NCI-H1975, HCC827, and NCI-H1299 cell lines were cultured in RPMI-1640 (Gibco, Life Technologies, NY, USA). The other cell lines were cultured in Dulbecco’s Modified Eagle’s Medium (DMEM). All the cells were supplemented with 100 μg/mL streptomycin, 100 U/mL penicillin (Beyotime, Shanghai, China), and 10% fetal-bovine serum (FBS) (TransGen Biotech, Beijing, China) at 37 °C in a humidified atmosphere of 5% CO_2_.

For cell transfection, the plasmids pcDNA3.1-Flag-NEDD4L mutants (ΔC2:193-975aa, ΔHECT:1-640aa, and ΔCW:581-975aa), pcDNA3.1-Flag-NEDD4L WT, pcDNA3.1-Flag-NEDD4L T367A, and pcDNA3.1-HA-EGFR WT and all related mutation plasmids (K716R, K737R, K754R, K860R, K867R, K929R, K970R) were constructed by YouBio (Changsha, China). The different mutations were constructed by site-directed mutagenesis using the QuickChange Site-Directed Mutagenesis Kit (Agilent Technologies, Waltham, MA, USA). The details of shJWA, Flag-JWA, and the corresponding plasmids have been described in a previous study [29]. Small interfering RNAs targeting AMPK, BCLAF1, CTBP1, HDAC2, ZMYM3, BCLAF1, FOXO3, and SMC3 were designed and synthesized by RiboBio (Guangzhou, China). The sequences are listed in Appendix A. The plasmids and siRNAs were transfected into cells using Lipofectamine 3000 (Invitrogen, Grand Island, NY, USA) according to the manufacturer’s guidelines. For the cycloheximide-chase experiments, the cells were treated with 100 μg/mL cycloheximide (CHX) (Selleck Chemicals, Shanghai, China) for different time points. For the ubiquitination assay, the cells were treated with 10 μM MG132 (Selleck Chemicals, Houston, TX, USA) for 6 h. AMPK-selective inhibitor Dorsomorphin (HY-13418A) and EGFR inhibitor osimertinib (AZD9291, HY-15772) were purchased from MedChemExpress (MCE).

### 4.2. CRISPR/Cas 9-Mediated Deletion of CTBP1

The sgRNAs were designed by the CRISPR sgRNA-design web tool as described previously [68]. To generate A549 CTBP1-knockout (CTBP1-KO) cells, The CTBP1 sgRNA sequences were designed as follows: sgRNA#1: AGGGGCCCGTTCATGATCGG, sgRNA#2: AGTGGCCACGTCCTTCAGGA, and sgRNA#3: CTGCGACGCGCAGTCCACGC. Cas9 protein and sgRNA were cloned into a plasmid. Briefly, after transfection for 48 h, 1 μg/mL puromycin (Sigma-Aldrich, St. Louis, MO, USA) was added to the cell-culture medium for another 7 days to selected transfected cells, followed by selection of individual clones via serial dilution. Then, the knockout of CTBP1 was verified by Sanger sequencing and Western blotting.

### 4.3. Compounds and Synthesis

JAC1 (C_15_H_10_FN_3_OS_2_, MW: 331.4), JAC4 (C_16_H_13_N_3_O_3_S, MW: 327.4), JAC4-R, and JAC4-S types were synthesized by Jiangsu Simcere Pharmaceutical Co., Ltd. (Nanjing, China). The purity of compounds was over 98% and was confirmed by HPLC-MS analysis. DMSO (Sigma-Aldrich, D2650) was used as solvent in vitro. The solution in vivo was prepared with polyethylene glycol (40%), ethanol (7.5%), and physiological saline (52.5%) by volume, and the vehicle was a solution only without compound.

### 4.4. RNA Isolation and Quantitative Reverse-Transcriptase Real-Time PCR

Total cellular RNA was extracted by TRIzol (Invitrogen, Carlsbad, CA, USA). We reverse transcribed RNA to cDNA using a reverse-transcription kit (Vazyme, Nanjing, China). Quantitative real-time RT-PCR was performed with the SYBR Green Master Mix Kit (Vazyme, Nanjing, China) on an Applied Biosystems 7900HT sequence-detection system (Applied Biosystems, Waltham, MA, USA). The primer sequences are shown in Appendix A.

### 4.5. Colony Formation and EDU-Staining Assays

For the colony-formation assay, 1 × 10^3^ lung-cancer cells were seeded into six-well plates, and 24 h later the cells were treated with JAC4 at the indicated doses and cultured for 10–14 days. The medium was changed every 3 days, then fixed with 4% paraformaldehyde and stained with crystal violet (Beyotime, Shanghai, China). Visualize colonies were counted. For EdU-staining assays, proliferating cells were detected with the BeyoClick^TM^ EDU Cell Proliferation Kit with Alexa Fluor 555 (Beyotime, Shanghai, China) according to the manufacturer’s protocol. Cells were seeded in 96-well plates and incubated for 2 h with EDU working solution (20 μM) after the cells had returned to the normal state, followed by fixation with 4% paraformaldehyde for 30 min. Then, they were incubated with PBS containing 0.3% Triton X-100 for 15 min at room temperature. To detect the percentage of cell proliferation, the cell nuclei were stained for 10 min using 2-(4-Amidinophenyl)-6-indolecarbamidine dihydrochloride (DAPI). Next, images of cells were acquired with a Nikon Ti microscope (Nikon, Tokyo, Japan).

### 4.6. Transwell Assay

Transwell cell-migration assays were performed in 24-well plates using 8 μM-pore polycarbonate-membrane inserts (Corning, Tewksbury, MA, USA). The bottom of the upper chamber was coated with fibronectin (Merck Millipore, Darmstadt, Germany); for the tumor-invasion assay, the chamber membranes were coated with 50 μL Matrigel (BD Biosciences, San Jose, CA, USA). After 48 h of transfection or JAC4 treatment, A549 cells or SPCA1 cells (2 × 10^4^) were seeded in the upper chamber on serum-free medium and the lower chamber was added to DMEM containing 10% FBS. After incubation at 37 °C for 12 h, cells were fixed with 4% paraformaldehyde, stained with crystal-violet solution, and counted at a magnification × 200 under a microscope.

### 4.7. Western Blot, Co-Immunoprecipitation (Co-IP), and Ubiquitination Assays

Cell samples were lysed with RIPA lysis buffer (1% Triton X-100, 1% sodium deoxycholate, 0.1% SDS, 50 mM Tris, 150 mM NaCl, pH 7.4) supplemented with protease inhibitors and phosphatase inhibitors (NCM Biotech, Suzhou, China). Protein concentrations were quantified using the BCA Protein Assay Kit (Beyotime, Shanghai, China). Then, proteins were separated by SDS-polyacrylamide gel electrophoresis (SDS-PAGE) and transferred to polyvinylidene-fluoride (PVDF) membranes (Millipore, Darmstadt, Germany), which were blocked by 5% nonfat milk in TBST for 1 h at room temperature and incubated with primary antibody overnight at 4 °C. After 4 washes with PBST, the membrane was incubated with horseradish-peroxidase (HRP)-conjugated secondary antibody for 1 h at room temperature. The primary and secondary antibodies used in the study are listed in Appendix A. The signals were visualized by an enhanced chemiluminescence (ECL)-detection kit (Vazyme, Nanjing, China).

For immunoprecipitation, the cells were lysed with NP40 lysis buffer (50 mM Tris-HCl, 150 mM NaCl, 1% NP40, 0.5% deoxycholate, pH 8.0) supplemented with a protease-inhibitor cocktail. Immunoprecipitation was performed using the indicated primary antibodies and protein A/G agarose beads (Santa Cruz, CA, USA) at 4 °C. The beads were washed 3 times with IP lysis buffer for 5 min each and the proteins were eluted by adding 2 × loading buffer (Beyotime) and boiling for 5 min. The immunoprecipitates were subjected to standard Western-blot analysis. For the ubiquitination assay, the cells were treated with DMSO or JAC4 for 24 h followed by treating with MG132 or not for 6 h (10 μM). Then, cells were harvested and protein samples were prepared and used for Western blotting and Co-IP assays, respectively.

### 4.8. Immunohistochemistry (IHC) and Immunofluorescence (IF) Staining

We performed immunohistochemistry and hematoxylin/eosin staining (H&E) on formalin-fixed paraffin-embedded (FFPE) tissue using standard protocols. The following antibodies were used for IHC: Ki67 (Abcam, Shanghai, China, ab15580, 1:250), NEDD4L (CST, #4013, 1:200), and cleaved caspase 3 (CST, #9664, 1:500). The details of IF assays were described previously [69]. Briefly, JAC4-treated lung-cancer cells were fixed with 4% paraformaldehyde for 1 h and then washed with PBST for 15 min, and non-specific sites were blocked with normal goat serum (Beyotime) for 1 h. Subsequent incubation with anti-JWA antibody (our laboratory, 1:100) and anti-EGFR antibody (Proteintech, Wuhan, China, 1:200) was carried out at 4 °C overnight and then with corresponding Alexa Fluor-labelled secondary antibodies (Beyotime, 1:200) for 1 h at room temperature. Next, cells were incubated with DAPI (Beyotime) for 10 min. Confocal images of stained cells were captured using Zeiss Aim software Version 2.3 on a Zeiss LSM 700 confocal microscope system.

### 4.9. Protein Half-Life Assays

The cells were treated with 10 μM JAC4 or DMSO for 24 h, then with CHX (100 μg/mL) for the indicated time periods. Cell lysates were collected for Western-blot analysis for EGFR protein levels using anti-EGFR antibody (CST, #4267, 1:1000).

### 4.10. Cellular Thermal-Shift Assay (CETSA)

The CETSA was performed by standard protocols [70]. HBE cells were treated with 10 μM JAC4 or DMSO for 6 h. Cells were suspended in phosphate-buffered saline containing protease inhibitors, heated at the indicated temperature for 3 min, and then immediately snap-frozen using liquid nitrogen. The samples were collected and subjected to Western-blot analysis.

### 4.11. Nuclear- and Cytosolic-Protein Extraction

Cells were extracted according to the instructions for the Nuclear and Cytoplasmic Extraction Kit (Beyotime, P0028). Briefly, A549 cells were treated with different concentrations of JAC4 for 24 h, after which the cells were collected by centrifugation and 200 µL of Cell Pulp Protein Extraction Reagent A were added per 20 µL of cell precipitate, shaken vigorously for 5 s, and placed in an ice bath for 10–15 min. Then, 10 μL of Cell Plasma Protein Extraction Reagent B were added, and the cells were shaken vigorously for 5 s and placed in ice bath for 1 min. They were then centrifuged at 12,000–16,000× *g* for 5 min and the supernatant was aspirated as the cytosolic protein. For the precipitate, 50 µL of nucleoprotein-extraction reagent were added and the supernatant was aspirated for nuclear protein by centrifugation after several rounds of vigorous shaking and an ice bath.

### 4.12. Molecular-Docking Assay

Three-dimensional protein structures of NEDD4L and EGFR were predicted using the bioinformatics tool I-TASSER (Iterative Threading Assembly Refinement), and protein molecular-docking predictions were made using the Discovery Studio 3.0 server, i.e., NEDD4L and EGFR. Local servers were used for docking-data-file processing and embellishment. The HDOCK server (http://hdock.phys.hust.edu.cn/, accessed on 12 October 2022) is a protein–protein-docking method based on a hybrid algorithm of template-based modeling and template-free docking [71]. The complex structures of AMPK-NEDD4L were predicted using the HDOCK algorithm based on the structures of AMPK and NEDD4L (Protein Data Bank structures).

### 4.13. Biotin-Assisted Pull-Down Assay and Mass-Spectrometry Analysis

Biotin-assisted pull-down assay and mass-spectrometry analysis were performed as described [72]. Cells were lysed in IP lysis buffer (20 mM Tris (pH 7.5), 150 mM NaCl, 1% Triton X-100, protease inhibitor, and 1 mM EDTA) and subjected to protein quantification. Briefly, whole-cell protein lysates (500 μg) were incubated with 10 μM biotin or 10 μM biotin-JAC4 for 6 h, followed by a 12 h incubation with streptavidin-coupled beads (Thermo Fisher Scientific, 65601) at 4 °C. Afterwards, biotin-bound beads and biotin-JAC4-bound beads were carefully washed by the eluent and then subjected to SDS-PAGE, the gels were stained with staining solution (Coomassie Brilliant Blue R-250 dye, Beyotime, P0017F) for 30 min and decolorized, and then the gels were cut carefully and analyzed by LC-MS/MS.

### 4.14. In-Vivo Xenograft Assay and Lung-Metastasis Assay

Six-week-old male BALB/c nude mice were purchased from the JiangSu Jicui pharmaceutical company and maintained in specific pathogen-free facilities. Tumor-xenograft and -metastasis models were created with the nude mice. These lung-tumor models were used in this project. (i) For xenograft models, 5 × 10^6^ human-lung-cancer SPCA1 cells suspended in 100 μL of PBS in the logarithmic growth phase were subcutaneously injected into the right flank of nude mice. When the average tumor volume reached 100 mm^3^, different treatment groups were intragastrically administrated: vehicle, JAC4 (100 mg/kg), and JAC1 (100 mg/kg), and tumor growth was measured every 2 days using sliding calipers. Tumor volume was calculated using the following formula: tumor volume = 1/2 length × width × width. The tumor was then removed, weighted, and frozen in liquid nitrogen for further analysis. (ii) For EGFR-mutant-cell xenograft models, 3 × 10^6^ NCI-H1975 (EGFR T790M) cells suspended in 100 μL of PBS were subcutaneously injected into the right flank of nude mice. When the average tumor volume reached 100 mm^3^, different treatment groups were intragastrically administrated: vehicle, JAC4 (100 mg/kg), AZD9291 (5 mg/kg), and a combination of JAC4 and AZD9291. On day 15, the mice were euthanized and the tumors were removed for subsequent analysis. (iii) For EGFR-mutant-cell lung-metastasis models, NCI-H1975 cells (1 × 10^6^) were injected into mice through the tail vein, and the mice were treated daily with different treatments starting the following day. At the end of experiment, the mice were sacrificed by over-anesthesia, and lung tissue was collected for histological analysis. All animal studies were approved by the Institutional Animal Care and Use Committee of Nanjing Medical University (IACUC-2012059).

### 4.15. Statistical Analysis

All data were expressed as mean ± SD. Student’s *t*-tests were used to compare means between two groups and ANOVA was used for comparisons among more than two groups. The X^2^ test was used to analyze the relationship between JWA expression and various clinicopathologic characteristics. Survival analysis was performed by Kaplan–Meier analysis with the log-rank test. Correlation between JWA expression and other gene expression was performed by the Spearman rank-correlation test. A *p* < 0.05 was considered statistically significantly. Additional experimental methods are described in the Appendix A.

## 5. Conclusions

In the present study, we provide evidence for the first time that JAC4 as an agonist of the JWA gene effectively inhibits the proliferation and metastasis of NSCLC. Mechanistically, JAC4 reduced the PI3K/AKT overactivation caused by both EGFR overexpression and mutations in NSCLC; JAC4 rescued JWA transcription in lung-cancer cells by binding to CTBP1. In addition, JAC4 phosphorylated and stabilized NEDD4L by JWA-triggered activation of the AMPK-signaling pathway; the phosphorylation of NEDD4L further accelerated the degradation of EGFR through enhanced ubiquitination at K716, therefore suppressing the progression of EGFR-driven lung cancer. Our results may provide a new strategy for EGFR-driven cancer therapy, especially for EGFR-mutant cancers.

## Figures and Tables

**Figure 1 ijms-24-08794-f001:**
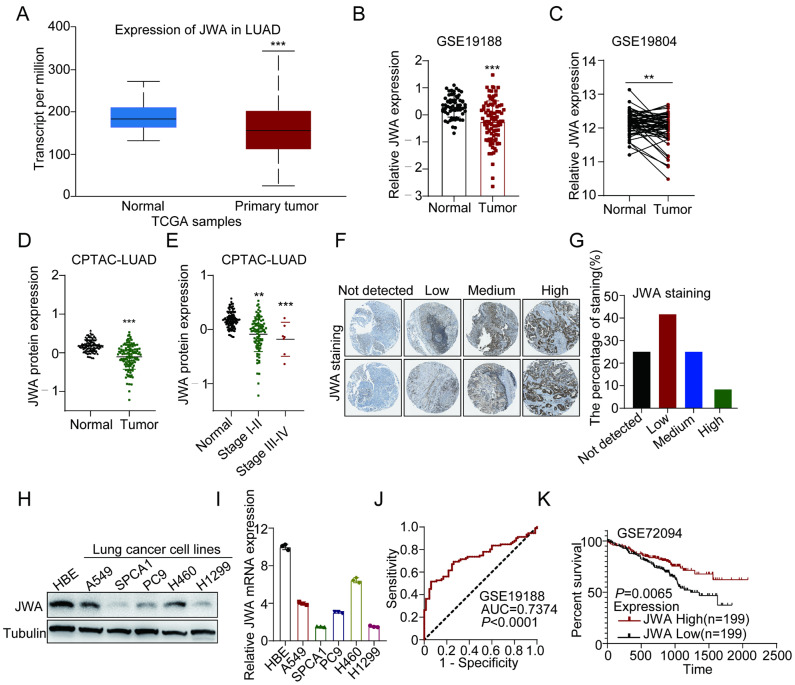
JWA is positively correlated with better outcomes for patients with lung cancer. (**A**) JWA expression in lung-cancer tissues (n = 515) and normal tissues (n = 59) from the TCGA database using the UALCAN (http://ualcan.path.uab.edu/analysis.html, accessed on 22 November 2021). (**B**) JWA is downregulated in lung cancer (n = 91) compared to adjacent normal lung tissues (n = 65) in the GSE19188 dataset. (**C**) JWA expression in paired lung-cancer and normal adjacent tissues (n = 60) from the GSE19804 dataset. (**D**,**E**) Protein expression of JWA in LUAD (N = 101, T = 110), different stages of LUAD (N = 101, stage I–II: n = 88, Stage III–IV: n = 22) from CPTAC. (**F**,**G**) Representative IHC images and analysis based on the database of The Human Protein Atlas (https://www.proteinatlas.org/, accessed on 20 January 2022). (**H**) The protein expression of JWA in NSCLC cells (A549, SPCA1, PC9, H460, H1299) and corresponding normal human bronchial epithelial (HBE) cells; three independent replicates were carried out. (**I**) Relative JWA mRNA expression in NSCLC cells compared to HBE cells, as determined by qRT-PCR. (**J**) ROC plot showing the AUC of the JWA expression from GSE19188. (**K**) Kaplan–Meier survival curves of overall survival (OS) based on JWA expression from the GSE72094 dataset. Data are presented as mean ± SD. ** *p* < 0.01, *** *p* < 0.001.

**Figure 2 ijms-24-08794-f002:**
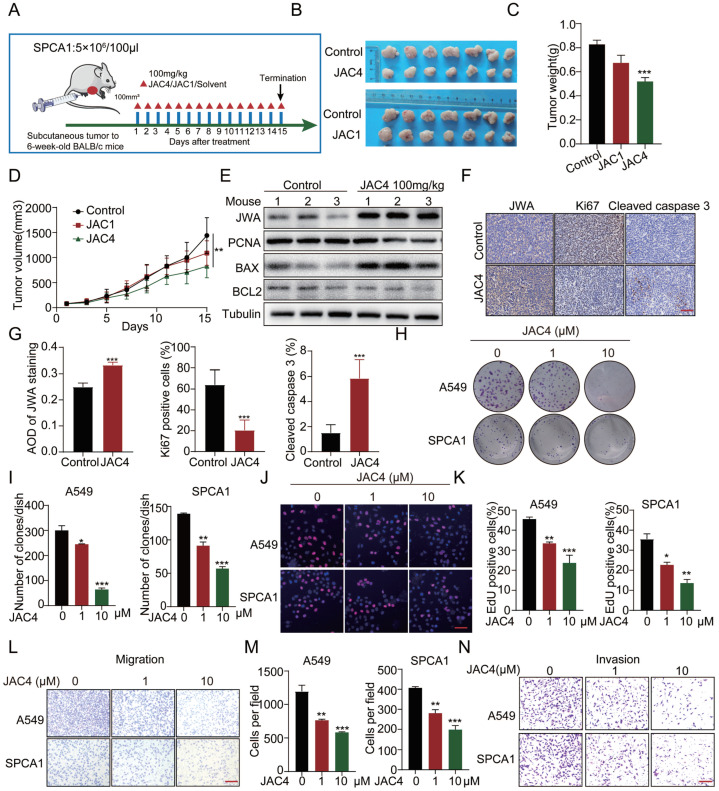
Screening of small-molecule compound agonist JAC4 of JWA gene for suppression of LUAD tumor growth. (**A**) In-vivo schematic showing a therapeutic-mouse model of lung-cancer tumor-bearing derived from the SPCA1 cell line. (**B**) Representative xenograft tumor image, (**C**) tumor masses, and (**D**) and tumor volumes for the indicated groups (n = 7). Scale bars = 50 μM. (**E**) Relative protein levels of JWA, PCNA, BAX, and BCL2 were detected in subcutaneous xenograft tumors by Western blot, n = 3. (**F**) Representative IHC image of JWA, Ki67, and cleaved caspase 3 immunostaining from xenograft tumors. (**G**) Quantification of the number of cells positive for Ki67 and cleaved caspase 3 staining; the average optical density (AOD) of JWA staining per field of vision in tumor sections (n = 3, 2 views per section). (**H**,**I**) The colony formation of A549 and SPCA1 cells after treatment with DMSO or JAC4; three independent replicates were carried out. (**J**,**K**) Assessment of the proliferation of A549 and SPCA1 cells after treatment with DMSO and JAC4 by EDU assay. EDU incorporation (red), DAPI-stained nuclei (blue). Three independent replicates were carried out. Scale bars = 50 μM. (**L**–**N**) Assessment of the migration and invasion of A549 and SPCA1 cells after DMSO or JAC4 treated by transwell assay; three independent replicates were carried out. Scale bars = 50 μM. Data are presented as mean ± SD. * *p* < 0.05, ** *p* < 0.01, *** *p* < 0.001.

**Figure 3 ijms-24-08794-f003:**
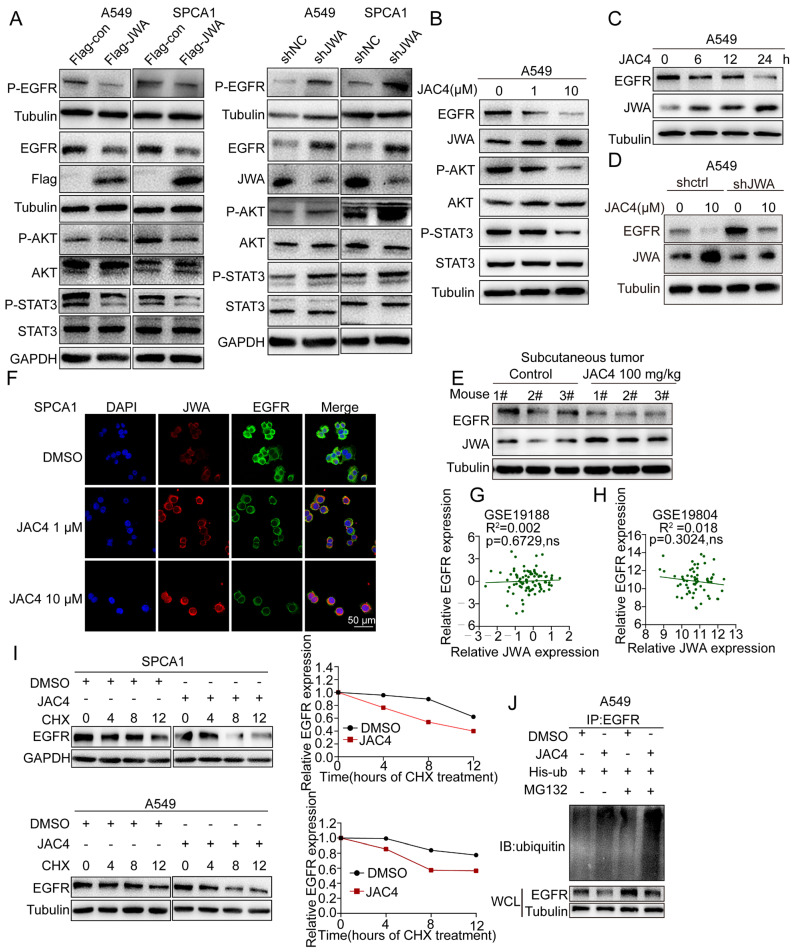
JAC4 modulates EGFR degradation through the ubiquitination–proteasome pathway. (**A**) Overexpression of JWA inhibited EGFR and its downstream markers; however, knockdown of JWA promoted EGFR expression and related markers. (**B**) A549 cells were treated with different concentrations of JAC4 for 24 h, and then JWA, EGFR protein expressions, and related markers were detected by Western blot. (**C**) Detection of JWA and EGFR protein expressions in A549 cells treated with JAC4 at different times. (**D**) JAC4-mediated EGFR degradation was blocked by silencing JWA in A549 cells. (**E**) Immunoblot analysis (IB) of JWA and EGFR expressions in tumor lysates of vehicle- and JAC4-treated mice (n = 3). (**F**) Immunofluorescence (IF) analysis of JWA and EGFR expressions in A549 treated with DMSO or JAC4, n = 3. Scale bars = 50 μM. (**G**,**H**) No correlation between JWA and EGFR mRNA expressions in lung-cancer tissues from the GSE19188 (**G**) and GSE19804 (**H**) data set. (**I**) A549 and SPCA1 cells were treated with DMSO or JAC4 (10 μM) for 24 h, followed by cycloheximide (CHX,100 μg/mL) treatment. Cells were harvested at the indicated time points and EGFR levels were determined by Western blot. EGFR protein-degradation kinetic curves based on the quantification of EGFR levels. (**J**) IB analysis of WCL and EGFR IP products derived from A549 cells transfected with indicated plasmids. Then, cells were treated with DMSO or JAC4 for 24 h, followed by incubation with or without MG132 (10 μM) for 6 h. Data are presented as mean ± SD. ns: not significant.

**Figure 4 ijms-24-08794-f004:**
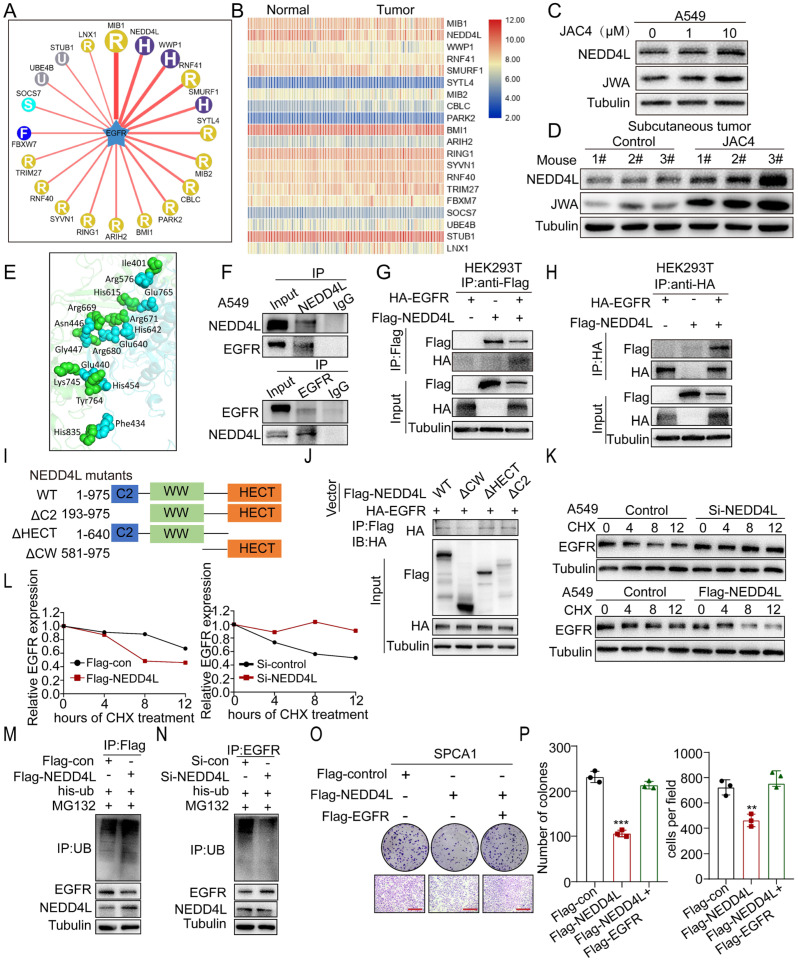
JAC4 ubiquitinates EGFR by E3 ubiquitin ligase NEDD4L. (**A**) The potential E3 ubiquitin ligase of EGFR using online bioinformatics UbiBrowser confidence mode (http://ubibrowser.ncpsb.org, accessed on 22 December 2021). (**B**) Expression levels of E3 ubiquitin ligase in 60 pairs of lung-cancer microarray data based on GSE19804. (**C**) IB analysis of NEDD4L and JWA protein levels in A549 cells treated with DMSO or JAC4 and (**D**) tumor lysates from a SPCA1 tumor-bearing model. (**E**) Molecular-docking assays between NEDD4L and EGFR. (**F**) A549 cells were pre-treated with MG132 (10 μM) for 6 h, and the endogenous protein–protein interaction between NEDD4L and EGFR was assessed by IP with anti-EGFR antibody or anti-NEDD4L antibody, detected by Western blot. (**G**,**H**) Exogeneous-protein interactions were demonstrated in HEK293T between NEDD4L and EGFR. (**I**) Schematic diagram of NEDD4L domains. (**J**) Co-IP analysis of the interaction of HA-EGFR with the indicated NEDD4L constructs in HEK293T cells. (**K**,**L**) A549 cells transfected with Flag-NEDD4L plasmids or si-NEDD4L were treated with CHX (100 μg/mL) prior to IB analysis. EGFR protein levels were quantified by normalization to tubulin. (**M**,**N**) In-vitro ubiquitination-assay analysis of NEDD4L expression in the ubiquitination of EGFR. (**O**,**P**) The colony formation and migration of SPCA1 cells transfected with indicated plasmids. Scale bars = 50 μM. Data are presented as mean ± SD (n = 3 independent experiments). ** *p* < 0.01, *** *p* < 0.001.

**Figure 5 ijms-24-08794-f005:**
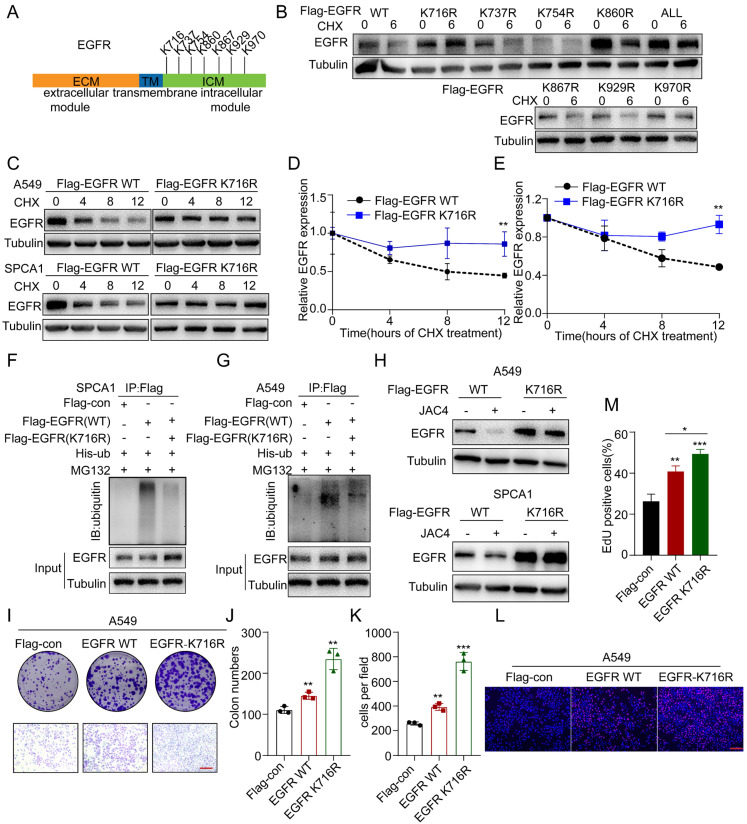
K716 of EGFR is critical for NEDD4L-mediated ubiquitination and degradation by JAC4. (**A**) Schematic of lysine mutants of EGFR. (**B**) The indicated plasmids of EGFR lysine mutations were transfected into A549 cells for 48 h, followed by exposure to CHX (100 μg/mL) for 6 h. Cells were lysed and immunoblotted with the indicated antibodies. (**C**–**E**) A549 and SPCA1 cells infected with Flag-EGFR (WT) or Flag-EGFR (K716R) plasmids were treated with CHX (100 μg/mL) prior to IB analysis. EGFR protein levels were quantified by normalization to tubulin. (**F**,**G**) A549 and SPCA1 cells transfected with his-ub were further transfected with Flag-control, Flag-EGFR (WT), or Flag-EGFR (K716R) for 48 h; then, the cells were treated with 10 μM MG132 for 6 h before co-IP and IB analysis. (**H**) A549 and SPCA1 cells were transfected with Flag-EGFR (WT) or Flag-EGFR (K716R) plasmids for 48 h, then treated with DMSO or JAC4 (10 μM) for 24 h before IB analysis. (**I**–**M**) A549 cells were transfected with Flag-control, Flag-EGFR (WT), or Flag-EGFR (K716R) plasmids and colony formation, transwell, (**I**–**K**), and EDU assays (**L**,**M**) were conducted, respectively. EDU incorporation (red); DAPI-stained nuclei (blue). Scale bars = 50 μM. Data are presented as mean ± SD (n = 3 independent experiments). * *p* < 0.05, ** *p* < 0.01, *** *p* < 0.001.

**Figure 6 ijms-24-08794-f006:**
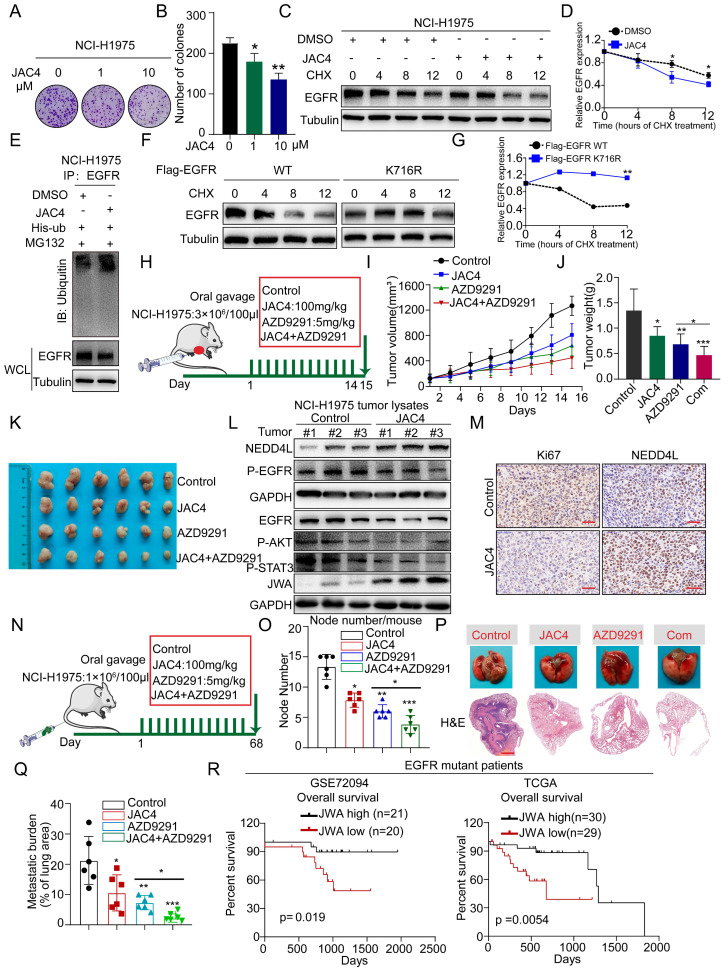
JAC4 suppresses EGFR T790M-driven LUAD growth and metastasis. (**A**,**B**) The colony-formation activity of NCI-H1975 cells was determined after treatment with different concentrations of JAC4. (**C**,**D**) NCI-H1975 cells were treated with DMSO or JAC4 (10 μM) for 24 h, then treated with CHX (100 μg/mL) prior to IB analysis. EGFR protein levels were quantified by normalization to tubulin (n = 3 independent experiments). (**E**) NCI-H1975 cells transfected with his-ub were treated with DMSO or JAC4 for 24 h, then treated with 10 μM MG132 for 6 h before Co-IP and IB analysis. (**F**,**G**) NCI-H1975 cells were transfected with Flag-EGFR (WT) or Flag-EGFR (K716R) for 48 h, then treated with CHX (100 μg/mL) prior to IB analysis. EGFR protein levels were quantified by normalization to tubulin (n = 3 independent experiments). (**H**) Schematic diagram of the establishment of the tumor-bearing model of EGFR mutant cells. (**I**) Tumor volumes, (**J**) tumor weight, and (**K**) image of tumor mass after treatment for 15 days (n = 6). (**L**) IB analysis of JWA, NEDD4L, EGFR, P-EGFR, P-AKT, and P-STAT3 expressions in tumor lysates from vehicle- and JAC4-treated mice, n = 3. (**M**) IHC analysis of Ki67 and NEDD4L expression in tumors from vehicle- and JAC4-treated mice. Scale bars = 100 μM. (**N**) Schematic representation of the in-vivo treatment model derived from the NCI-H1975 cell line for metastatic EGFR-mutant lung cancer (n = 6). (**O**) Lung metastatic nodes, (**P**) representative H&E staining of lungs, and (**Q**) lung metastatic burden determined from histology of EGFR-mutant lung-cancer-metastasis model. Scale bars = 2 mm. (**R**) Kaplan–Meier survival curves of overall survival (OS) based on JWA expression in EGFR-mutant lung-cancer patients from GSE72094 and TCGA datasets. Data are presented as mean ± SD. * *p* < 0.05, ** *p* < 0.01, *** *p* < 0.001.

**Figure 7 ijms-24-08794-f007:**
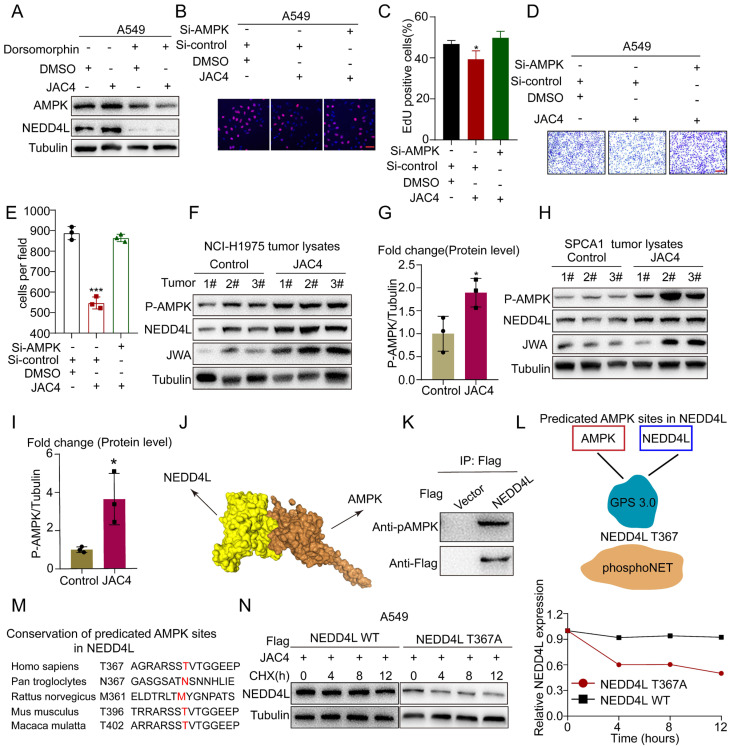
JWA/AMPK axis stabilizes NEDD4L expression by phosphorylating NEDD4L at Thr367. (**A**) IB analysis of A549 cells that were treated with 10 μM JAC4 for 24 h with or without AMPK inhibition. (**B**,**C**) The proliferation and migration of A549 cells were treated with 10 μM JAC4 for 24 h with or without AMPK knockdown by EDU assay, and (**D**,**E**) transwell assay. EDU incorporation (red), DAPI-stained nuclei (blue), n = 3 independent experiments. (**F**,**G**) IB analysis of protein levels in NCI-H1975 tumor lysates, n = 3. (**H**,**I**) IB analysis of protein levels in SPCA1 tumor lysates, n = 3. (**J**) Molecular docking between NEDD4L and AMPK was determined via the HDCOK server (http://hdock.phys.hust.edu.cn, accessed on 12 October 2022). (**K**) Co-IP showing that Flag-NEDD4L interacted with the activated form of AMPK (AMPK pT172). (**L**) The prediction of the putative phosphorylation site of NEDD4L by AMPK was through a combination of the GPS 3.0 software and the PhosphoNET kinase predictor. (**M**) Conservation of predicted of AMPK sites in NEDD4L was shown in different species. (**N**) A549 cells were transfected with Flag-EGFR (WT) or Flag-EGFR (T367A) for 48 h and were then treated with CHX (100 μg/mL) prior to IB analysis. EGFR protein levels were quantified by normalization to tubulin. Data are presented as mean ± SD (n = 3 independent experiments). * *p* < 0.05, *** *p* < 0.001.

**Figure 8 ijms-24-08794-f008:**
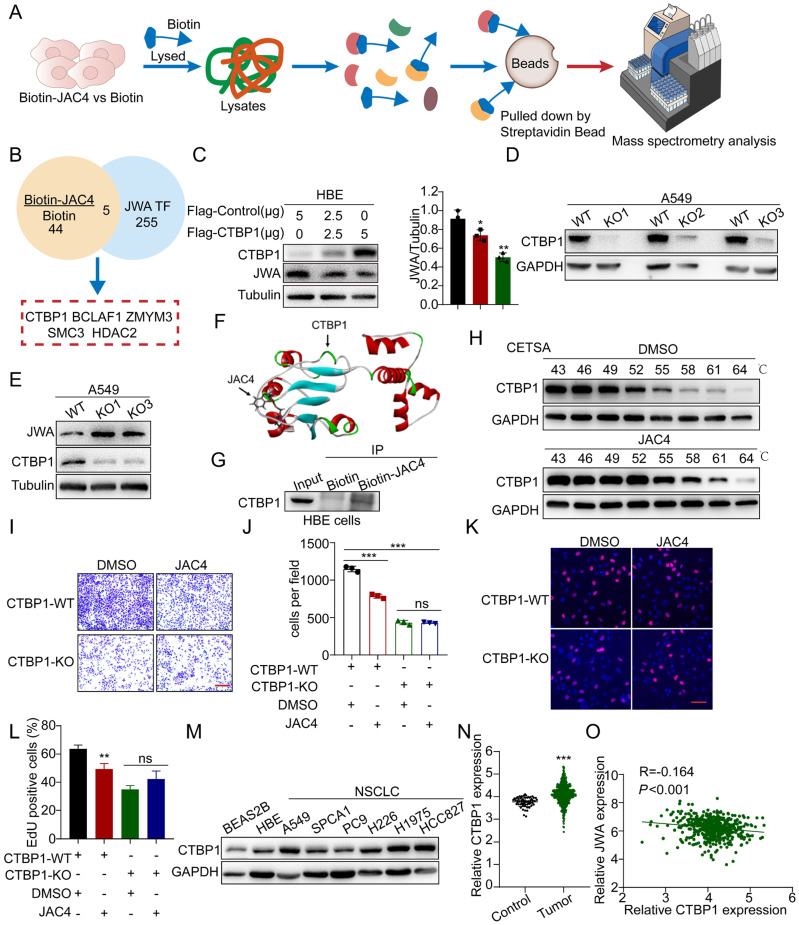
JAC4 upregulates JWA expression by binding to its transcriptional suppressor CTBP1. (**A**) Schematic showing the procedure of identifying JAC4 binding targets through mass spectrometry (MS). (**B**) Venn diagram showing that targets of JAC4 were predicted by MS and the JASPAR database. (**C**) HBE cells were transfected with Flag-control together with varying amounts of Flag-CTBP1 plasmids for 48 h, and then an IB analysis of JWA and CTBP1 expression was conducted; n = 3 independent experiments. (**D**) IB analysis of KO efficiency of CTBP1 by CRISPR/Cas9 in A549 cells. (**E**) KO of CTBP1 increased levels of JWA protein levels in A549 cells. (**F**) Molecular docking between JAC4 and CTBP1. (**G**) Co-IP showing that JAC4 interacted with CTBP1 in HBE cells. (**H**) CETSA was performed using HBE cells. (**I**,**J**) Assessment of the migration of A549 cells with or without CTBP1 KO treated with DMSO or 10 μM JAC4 by transwell assay; n = 3 independent experiments. (**K**,**L**) Assessment of the proliferation of A549 cells with or without CTBP1 KO treated with DMSO or 10 μM JAC4 by EDU assay, n = 3 independent experiments. EDU incorporation (red), DAPI-stained nuclei (blue). (**M**) IB analysis of CTBP1 protein expression in NSCLC cell lines compared to BEAS-2B and HBE cells. (**N**) Higher CTBP1 mRNA expression in lung-cancer tissues (TCGA data set) compared to corresponding control-lung tissues. (**O**) Negative correlation between JWA expression and CTBP1 expression from analysis of TCGA dataset. Data are presented as mean ± SD. * *p* < 0.05, ** *p* < 0.01, *** *p* < 0.001. ns: not significant.

**Figure 9 ijms-24-08794-f009:**
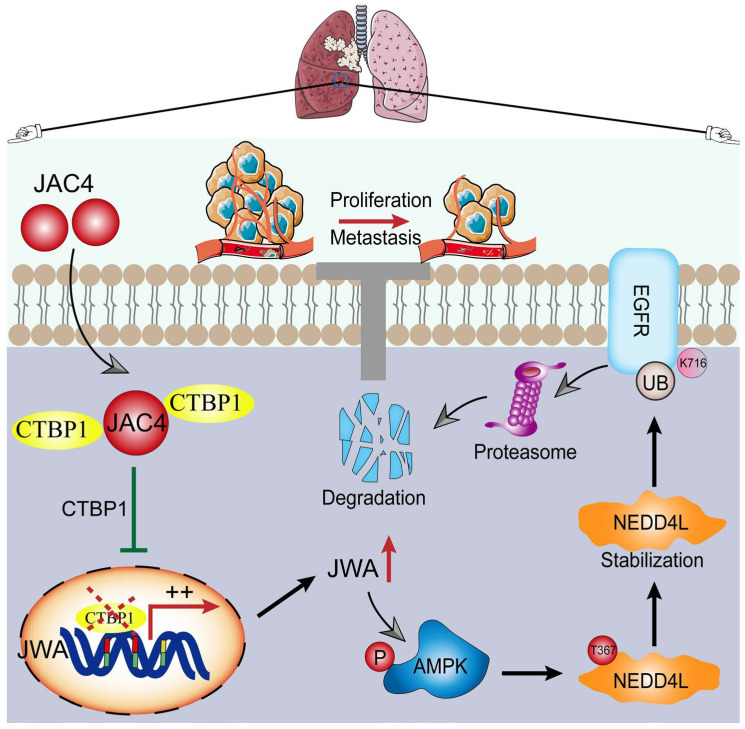
Schematic diagram of the mechanisms of JAC4 as a therapeutic agent on NSCLC. The small-molecular JWA gene agonist JAC4 plays a therapeutic role in EGFR-driven lung-cancer growth and metastasis through the AMPK/NEDD4L/EGFR axis.

## Data Availability

The data supporting the conclusions of this article are presented within the article and its additional files.

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
