# Peer review of "JAC4 Inhibits EGFR-Driven Lung Adenocarcinoma Growth and Metastasis through CTBP1-Mediated JWA/AMPK/NEDD4L/EGFR Axis"

_ijms, 2023, doi:10.3390/ijms24108794_

Round 1
Reviewer 1 Report
In the manuscript entitled “JAC4 inhibits EGFR-driven lung cancer growth and metastasis through CTBP1-mediated JWA/AMPK/NEDD4L/EGFR axis, Ding et al. showed a great amount of data supporting the idea that JAC4, a JWA agonist, could serve as a new therapeutic molecule in EGFR-driven lung cancer. Moreover, the authors elucidated the molecular mechanisms at the basis of JAC4 therapeutic activity. The study is overall novel and complete. However, there are important issues regarding some experimental analysis reported in the present manuscript that need to be addressed.
Major issues:
1) Figure 1F-G: the authors claimed that “The Human Protein Atlas data showed JWA was robustly reduced in lung cancer tissues”. This analysis is affected by important issues:
- no information about the cases included. The authors should provide a supplementary table of the cases included in the study;
- quantification in Figure 1G is lacking error bars and statistical test. Please provide these information otherwise the analysis is rather qualitative and do not support what is claimed in the text
- The same analysis was performed for NEDD4L protein. Please change it accordingly.
2) Figure 1 H: no information about the number of biological replicated performed and no statistical test. In this version the analysis is rather qualitative and do no support what is reported in the text.
3) Figure 1K: GSE72094 dataset analysis showed that patients with high JWA has better prognostic outcomes. Is this data confirmed in other datasets such as Human protein atlas or others?
4) Figure 2F: this analysis lacks quantification. Please provide it in the figure together with the number of cases analyzed.
5) The authors claimed that “Collectively, these findings suggest that JAC4 inhibits lung cancer progression in vivo through up-regulating JWA expression.” This is an overstatement since the data reported do not directly prove that increase in JWA expression is responsible for the reduction of tumor growth. Please change this sentence accordingly.
6) Figure 2J. the reprensentative image of Edu staining is at low resolution. Please provide an image at higher resolution to better depict the Edu staining (red). Please indicate for the reader that edu staining is in red in the relative legend. This problem is present also in other figures showing Edu staining.
7) Is JAC4 treatment able to inhibit also invasion in transwell assay?
8) The following sentence at paragraph 2.3 lacks some part. “On the basis of our previous findings that JWA suppresses cell migration by negatively regulating HER2 expression in HER2-positive gastric cancer cells[34] and JAC1 (JWA gene activating compound 1) inhibits the proliferation of HER2-positive breast can- cer cells through ubiquitination of HER2 [30].” Please revise it.
9) Figure 3A: The authors should complete the analysis by performing western blot for pEGFR to better check the activation of the pathway.
10) Figure 3D lacks the condition shNC+JAC4 and a western blot showing JWA regulation.
11) Figure 3F: This figure is affected by several issues. First, few cells are represented in the image (e.g. 2 to 4) and there is no quantification of the observed results. Moreover, some DAPI staining images are different between the single color and the merged images for the same condition (e.g. SPCA1 0, SPCA1 1, A549 0). This image should be strongly revised. Finally, in the text the authors claimed “immunofluorescence staining showed that the fluorescence intensity levels of JWA were increased, but the level of EGFR was decreased in tumor tissues”. However, this analysis was perfomed in cell lines. Please change it accordingly.
12) Figure 4: The experiments presented in this figure showed that NEDD4L expression is regulated by JAC4 treatment and is able to regulate the half-life of EGFR expression. To prove that JAC4 ubiquitinates EGFR by regulating NEDD4L as the author stated in the manuscript, ciclohexamide experiment in which cells are silenced with a siRNA against NEDD4L and treated with JAC4 should be performed.
13) Figure 4: Data on NEDD4L in tumors are already reported in literature (https://doi.org/10.1186/s12967-022-03247-4). Since this part of the story is not novel, these data should be moved in the supplementary. I would suggest to maintain in the main figure only the data about NEDD4L regulation by JAC4. Moreover, the paper should be cited and properly discussed, also because NEDD4L expression is reported to be downregulated by EGFR, that in turn is downregulated by JAC4 treament.
14) Figure 5B: EGFR protein (time 0) at basal levels is changing among the different conditions. Why? Is this western blot representative?
15) Figure 6L: the authors did not show the level of phosphorylated EGFR (e.g. active)? Is this reduced upon treatment with JAC4?
16) Figure 8O: Which kind of correlation was performed? Which is the R coefficient?
Minor issue:
The authors did not mention the number of biological replicates for each experiment in the figure legends. Please add it.
Author Response
Please see the attatchment.

Reviewer 2 Report
The paper showed the role of JAC4 in EGFR-driven lung cancer. Furthermore, the paper suggested related mechanisms that JAC4 increased the stability of NEDD4L through AMPK-mediated phosphorylation at Thr367. Sufficient data were presented to support the evidence and explain the mechanism in vitro and in vivo. There is clear scientific merit in NSCLC and the motivator to develop the manuscript. However, in its present stage, the manuscript presents deficiencies that should be revised to reach the possibility of publication.
- Do you have data for JWA staining by classifying LUAD and LUSC in Figure 1?
- Is there a correlation between JWA and NEDD4 mRNA expression in lung cancer tissues as shown in Figure 3G and H?
- In Figure 6M, the quantification of IHC of Ki67 and NEDD4L is required.
Author Response
Please see the attatchment.

Reviewer 3 Report
1) The selected cell line is A549 is indicating of lung adenocarcinoma. Thus, the title and whole of the manuscript content must be adjusted.
Lung cancer is a heterogenous cancer and the authors can not include these findings only by evaluation of A549 lung adenocarcinoma.
2) There are a wide range of lung cancer treatment options. It is strongly suggested that these agents add to the introduction using the relevant publications such as Jianwei Zhu, et al - 2021 , Fausto Petrelli, et al - 2021 , Mohammad Hadi Abbasian, et al - 2022 and so on..... . Then, the authors should clarify the rational behind this treatment strategy.
3) The quality of picture are low.
Author Response
Please see the attatchment.

Round 2
Reviewer 3 Report
No comments